# 'The illness isn't the end of the road'—Patient perspectives on the initiation of and early participation in a multi-disease, community-based exercise programme

Joanne Regan-Moriarty[1]*, Sarah Hardcastle[2], Maire McCallion[1], Azura Youell[1], Audrey Collery[3], Andrew McCarren[4], Niall Moyna[5]*, Brona Kehoe[6]

1 Department of Health and Nutritional Science, Atlantic Technological University, Sligo, Ireland, 2 Academy of Sport and Physical Activity, Sheffield Hallam University, Sheffield, United Kingdom, 3 Primary Care, Health Service Executive, Sligo, Ireland, 4 School of Computing, Dublin City University, Dublin, Ireland, 5 School of Health and Human Performance, Dublin City University, Dublin, Ireland, 6 Department of Sport and Exercise Science, South East Technological University, Waterford, Ireland

* joanne.regan@atu.ie (JRM); niall.moyna@dcu.ie (NM)

**Data Availability Statement:** All relevant data are within the manuscript and its Supporting Information files.

## Abstract

### Background

Exercise is the cornerstone of cardiac rehabilitation (CR). Hospital-based CR exercise programmes are a routine part of clinical care and are typically 6–12 weeks in duration. Following completion, physical activity levels of patients decline. Multi-disease, community-based exercise programmes (MCEP) are an efficient model that could play an important role in the long-term maintenance of positive health behaviours in individuals with cardiovascular disease (CVD) following their medically supervised programme.

### Aim

To explore patients experiences of the initiation and early participation in a MCEP programme and the dimensions that facilitate and hinder physical activity engagement.

### Methods

Individuals with established CVD who had completed hospital-based CR were referred to a MCEP. The programme consisted of twice weekly group exercise classes supervised by clinical exercise professionals. Those that completed (n = 31) an initial 10 weeks of the programme were invited to attend a focus group to discuss their experience. Focus groups were transcribed and analysed using reflexive thematic analysis.

### Results

Twenty-four (63% male, 65.5±6.12yrs) patients attended one of four focus groups. The main themes identified were 'Moving from fear to confidence', 'Drivers of engagement,' and 'Challenges to keeping it (exercise) up'.

**Funding:** The author(s) received no specific funding for this work.

**Competing interests:** The authors have declared that no competing interests exist.

## Conclusion

Participation in a MCEP by individuals with CVD could be viewed as a double-edged sword. Whilst the programme clearly provided an important transition from the clinical to the community setting, there were signs it may breed dependency and not effectively promote independent exercise. Another novel finding was the use of social comparison that provided favourable valuations of performance and increased exercise confidence.

## Introduction

In Ireland, over 268,000 live with cardiovascular disease (CVD), of which 67,000 have coronary artery disease [1]. With a growing and aging population [2], the prevalence of CVD is predicted to continue to rise [3], which will place a substantial burden on the Irish healthcare system. Physical activity is well established as an effective intervention in the secondary prevention of CVD [4]. It is the cornerstone of hospital-based cardiac rehabilitation (CR) programmes which are effective in the prevention of further cardiac events [3, 5]. Hospital-based CR programmes are part of routine clinical care delivered by health care professionals (HCP) but are short term, structured and of fixed duration (6–12 weeks) [6, 7]. They aim to '*improve the physical and emotional health and quality of life of patients*' [8] to facilitate the patient to resume an active and functional life [9] following a cardiac event. Following completion of hospital-based CR, physical activity levels can decline, with only 25–40% maintaining exercise six months post CR [10] and the health-related benefits achieved can be lost [11]. The uptake of long-term maintenance programmes (such as phase IV community-based CR (CBCR) programmes) could act as a vehicle to maintain exercise behaviour [12] and those that do transition into CBCR have been shown to have better health outcomes [11]. CBCR can provide a safe and effective exit route from hospital-based programmes [13]. The accessibility of community programmes and the potential for long term/continual service provision is likely to support the habituation of exercise and maintenance of associated health benefits [14]. Patients have expressed how the community setting promotes a sense of 'normality' within the rehabilitation experience, distinguishing exercise as a normal behaviour as opposed to a treatment for their condition [13]. However, it is estimated that only 5–20% of eligible CR patients are still attending maintenance/CBCR programmes [15] at 6 months.

Limited studies have focused on the facilitators of adherence to CBCR programmes [16–19]. These studies identified the social nature of the programme as a powerful motive for long-term exercise maintenance referring to the support given by instructors and exercising alongside people with similar health problems. Participants also valued the support and encouragement given by family and friends. The group class was a strong facilitator with many referring to the routine associated with a scheduled exercise time [18, 19], the novel exercises giving a new dimension to their physical activity [18] and the enjoyment of taking part in the class [16–19]. Another factor driving long-term exercise maintenance was the importance of being able to spend time with family, being able to travel, and being able maintain independence [19]. Ability to avoid ill health, indicating that health was perceived to be in their control, was also evident [16–19]. Lack of knowledge, lack of social support, poor health, and lack of medical support are described as key barriers to initiation of a CBCR [10]. Barriers to adherence include travel and lack of appropriate locations/time, other health problems, time constraints such as family/work commitments, and weather as key barriers [16, 20].

All these studies, however, involved participants that were attending for between 12 months [16, 18, 20] to 2+ years [17, 19], who were in true "maintenance" stage of change in terms of their exercise behaviour. There is value in understanding participants' motivations for and barriers to adhering to the exercise programmes at different time points [17]. Research into such programmes has consistently shown that dropout is highest within the first three months [21, 22].

Uptake to maintenance CBCR programmes is hindered by the inadequate availability of such services [23, 24] with the majority of these programmes delivered by HCPs such as physiotherapists [25, 26], GPs [27] or a multidisciplinary team including cardiologists, physiotherapist, CR nurses and exercise physiologists [16, 28]. Traditionally long-term exercise programmes have been delivered for individual CDs such as CR and pulmonary rehabilitation. However, the role of exercise as an adjunctive therapy in the management of a range of CDs is well documented [29]. As these rehabilitative programmes have comparable designs and target similar components of fitness irrespective of the CD [30], there is scope to establish multi-disease, community-based exercise programmes (MCEP), which would be a more efficient delivery model and could be used as an alternative to CBCR. No research to date has explored the perceptions of individuals with CVD integrating into a MCEP.

This present study aimed to explore patients experiences of the initiation and early participation in a MCEP programme and the dimensions that facilitate and hinder physical activity engagement in individuals with CVD.

## Methods

### Participants

Purposive sampling was employed [31] which included adults with established CVD who had completed hospital-based CR at Sligo University Hospital (SUH) who were referred to a MCEP at the Knocknarea Sports Arena, on the Atlantic Technological University (ATU) Sligo campus and were still attending at 10 weeks. Patients meeting the inclusion criteria for category B CVD of the National Exercise Referral Framework [32] (those who do not require the presence of a physician or other appropriately trained HCP i.e. advanced cardiac life support (ACLS) or equivalent, to undertake a supervised exercise programme) were referred by the senior cardiac physiotherapist at SUH. Some had completed their hospital-based CR prior to the MCEP being established but were subsequently referred once established, while others were referred directly following completion of their hospital-based CR. Participants were informed of the research study and provided with a plain language participant information leaflet. Participants were required to provide written informed consent prior to participation. The study conformed to the Standards for Reporting Qualitative Research [33] (S1 Appendix). Ethical approval was obtained from SUH Research Ethics Committee (REF No.: 579). A total of 51 cardiac patients were referred and commenced the MCEP between the 6th of June 2016 and the 19th of September 2017. At 10 weeks, 31 participants were still attending the MCEP and were invited to attend a focus group.

### Multi-disease, community-based exercise programme

The MCEP was ongoing and inducted new participants three times a year. Participants signed up for a 10-week block with the option to continue. The programme was offered to participants with a range of chronic diseases and primarily received referrals of patients with CVD, stroke, multiple sclerosis, Parkinson's disease, musculoskeletal, inflammatory bowel disease, and Type 2 Diabetes Mellitus.

**Table 1. Description of MCEP FITT parameters.**

| Parameter | Description |
|---|---|
| *Frequency (F)* | 2/week, supervised sessions |
| *Intensity (I)* | 'moderate' intensity, 11–14 on the 6–20-point Borg RPE scale, THR = 40–70% of HRR* |
| *Time (T)* | 60min |
| *Type (T)* | Aerobic exercise activities included stationary cycling, elliptical cross-training, rowing, treadmill walking, running, aerobic circuit stations and dance |
| | LME exercises involved circuit-based exercises using free weight or body resistance exercises |

BORG-RPE Borg Rate of perceived exertion, THR Training heart rate, HRR Heart rate reserve *subtract 30bpm if beta-blocked, LME Local muscular endurance

Exercise classes consisted of twice weekly 60-minute supervised group exercise delivered by clinical exercise professionals at a participant-to-instructor ratio of 15:1 [34]. Details of the exercise prescription are outlined in Table 1. At least one instructor was a Health and Exercise Scientist with level 4 cardiac exercise instructors' qualification from the British Association for Cardiovascular Prevention and Rehabilitation (BACPR). The classes consisted of a combination of aerobic and local muscular endurance (LME) exercises and included a 15 min warm-up and 10 min cool-down as per standard CR guidelines [34]. Participants completed a pre-exercise health check before each class and the physiotherapists were available on-site for the health check in the first 2–3 weeks of any newly inducted cohort and remotely thereafter. Classes finished with a social gathering with refreshments provided. Two educational workshops took place over each 10-week period delivered by a heath care or exercise professional. Topics covered included adopting a healthy lifestyle, the importance of taking their medication, practical nutrition advice.

Prior to starting the programme all participants underwent an induction where they completed a series of physiological tests measuring functional capacity, anthropometrics along with assessments of their health and wellbeing. Repeat assessments were performed at 10 weeks.

## Focus groups

Four focus groups with four to eight participants (mixed gender) per group and a single interview (due to work commitments) were conducted. Using a phenomenological approach, the focus groups aimed to explore participant opinions/experiences of participating in the first 10 weeks of a MCEP. A topic guide (S2 Appendix) was developed based on Braun and Clarke [31] to guide focus group discussions. Topics included the journey to MCEP, experience of the programme, the exercise class, factors that facilitated participation in the programme, perceived benefits of the MCEP setting and recommendations for exercise programme improvements. Discussions were not limited to these areas and opportunity was provided for the exploration of other/wider topics identified by participants. Sample size was determined by data saturation, the point when no new information was gained [31].

Focus groups and the interview were conducted by two trained independent researchers that had not been involved in the delivery of the MCEP. One acted as the moderator who introduced the session and followed the topic guide, while the second was the assistant moderator and manually recorded the key discussion points. Focus groups lasted approximately 45 min and were held in a meeting room in the exercise facility at ATU Sligo, the same building where the exercise classes took place to ensure the setting was familiar to the participants.

Focus groups were digitally recorded, transcribed verbatim and anonymised with each participant given a unique code.

## Authors experience with MCEPs

There was differing levels of experience with MCEPs between the key researchers who analysed the data. JRM had been part of establishing and operating the MCEP and has a background in exercise physiology. Due to her closeness to the day-to-day running of the programme she represented an 'insider' perspective [35] in the analysis of the data. MMcC represented an 'outsider' perspective [35] as she was not involved in MCEP though does have a strong background in public health/health promotion and qualitative research methods. In the later phases, BK and SH contributed to the analysis. BK has expertise in establishing and operating other MCEPs. SH has a wealth of experience and expertise in qualitative design and analysis. Neither BK or SH had direct involvement in the operation of the local programme or data collection, so both were deemed to contribute an 'outsider' perspective. Having both an insider and outsider perspective strengthened the study design [36] giving a more reflective and varied viewpoint during analysis.

## Data analysis

Data was inductively analysed following the six phases of reflexive thematic analysis as outlined by Braun and Clarke [37, 38]. In the first phase both researchers (JRM & MMcC) *familiarised* themselves with the transcripts by manually listening back to the audio tapes. In the second phase, transcripts were discussed jointly to *generate representative codes*. The third phase involved the generation of *Initial themes* followed by in-depth discussion between the researchers where they *reviewed and developed* the themes (Phase 4). At this point another perspective was brought in to give a fresh viewpoint (BK) before phase 5 took place, *refining, defining, and naming* of the themes to ensure all themes correctly reflected the transcripts. The thematic analysis report (phase 6) was drafted (JRM), discussed and revised (JRM, MMcC, BK, SH) with key quotations selected (JRM) for each theme.

## Results

Twenty-four (77%) of invited participants took part in the study (Fig 1). Characteristics of participants are presented in Table 2.

Three major themes were identified: 'Moving from Fear to Confidence', 'Drivers of Engagement,' and 'Challenges to Keeping it (Exercise) Up'. S3 Appendix provides an overview of themes, subthemes and supporting quotes.

## Moving from fear to confidence

This theme contained four sub-themes; Fear and uncertainty; Need for continuity; Increase in confidence; and Life beyond illness.

**Fear and uncertainty.**   Despite undertaking 10 weeks hospital-based CR prior to the MCEP many participants expressed fear and uncertainty in exercising independently. Participants knew they should be active but described feelings of fear or nervousness towards exercise *'Yes, fear is the thing, in case we overdo it.'* (P3 FG1) They were fearful of overexerting themselves, exacerbating their condition or inducing symptoms of their condition *'I was wondering would it bring on the pain, the angina I had'* (P6 FG1) and *'once this* [the cardiac event] *happened, I was afraid to walk.'* (F3 FG3) Many clearly recalled their period of ill health.

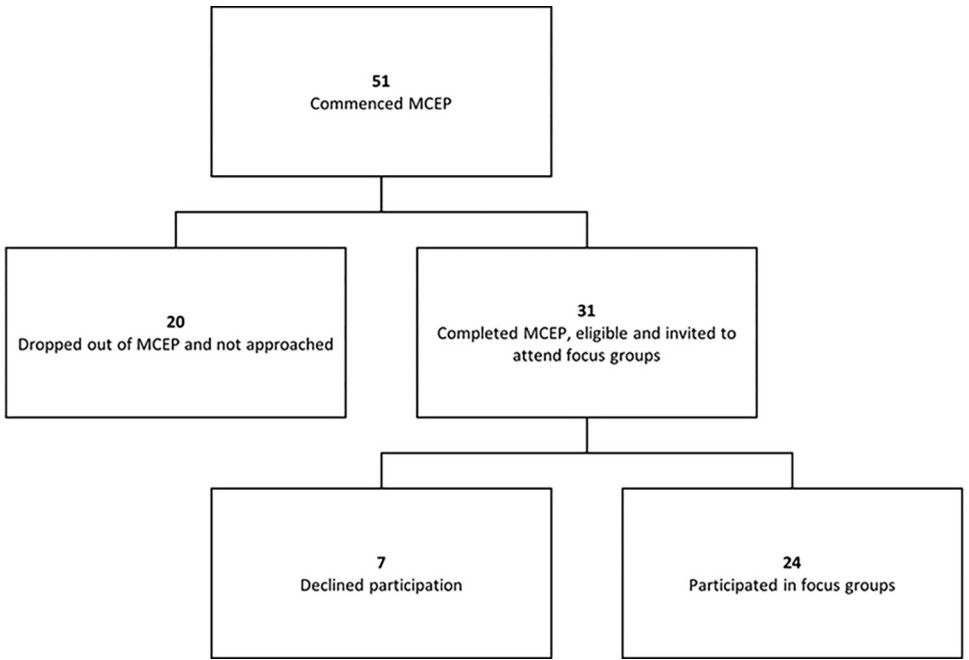

**Fig 1. Flowchart of eligible participants and recruitment.**

Participants were uncertain of how to exercise independently following the hospital-based programme: '*sure, I wouldn't know what I needed to do*' (M3 FG4). This manifested as low exercise confidence '*Well, I thought I mightn't be able to do much, or I wouldn't be able to manage it* [exercise].' (M1 FG4)

**Need for continuity.** There was a strong sense that participants wanted to maintain their exercise after hospital-based CR but needed support in doing so. There was a desire and perceived need for follow-on supervised exercise opportunities '*. . .it was something we were waiting for, all of us.*' (P1 FG1) Many were referred directly from the hospital-based CR to the MCEP and referenced the continuity of care. '*The continuity and the link between the hospital and yourselves* [the MCEP] *is critical, it's really important.*' (M3, FG4)

The access to medical support if they '*felt a wee pain*' or had a particular worry provided significant reassurance. Although not always on site, it was enough to know the medical support

**Table 2. Participant characteristics.**

| N | 24 |
|---|---|
| *Age (yr)* | 65.5±6.12 |
| *Gender, M:F* | 15:9 |
| *BMI (kg m$^2$)* | 29.3±4.7 |
| ***Cardiac History*** | |
| *Myocardial Infarction* | 13 (54%) |
| *Percutaneous Coronary Intervention (PCI) with stent* | 16 (67%) |
| *Coronary artery bypass graft (CABG)* | 4 (17%) |
| *Percutaneous Coronary Intervention (PCI) without stent* | 3 (13%) |
| *Valve Replacement* | 5 (21%) |

Values are expressed as absolute values (percentage) or mean ± standard deviation

was there if needed it '*We've got that link. We don't want to lose that support now that we have it.*' (M1, FG3) For some, this had been their second cardiac event and they commented '*there was nothing*' there after their first experience of CR in the hospital.

Although there was evidence of some being active without the MCEP, this was primarily walking or cycling. Many had never exercised in a gym previously or completed circuit training. The MCEP provided the knowledge, skill, and confidence to perform other activities.

'*You see those gym things, I would have never walked into a gym, . . .. you would feel really awkward going in, you wouldn't know what you were doing before that. At least now when I sit down, I know what I'm supposed to be doing.*' (P10, FG2)

**Increase in confidence.** The delivery of the MCEP by clinical exercise professionals, with referrals and support from HCPs, was an essential aspect to the programme for participants '*The combination of all of them was very beneficial.*' (P3, FG1) that lead to a sense of confidence and trust over other exercise opportunities. No participants reported any serious adverse events with participants perceiving the MCEP as a safe exercise environment '*you actually feel safe in the environment, which is supervised, they are professionals, and nothing is going to happen. But if something did happen, they are the right people to deal with it*' (P2, FG1)

The clinical exercise instructors were viewed positively '*. . .you trusted the staff that were looking after us, you knew that they were not going to put you in harm's way, they are there to take care of you.*' (P7 FG1) Participants recognised that the instructors gradually progressed the programme, instilling confidence they were exercising at an appropriate level. '*They do 3 stages* [demonstrate 3 options] *of what you can do yourself, you're not being pushed into working flat out, you do what you can.*' (M2 FG4)

Through participation in the MCEP, there was a noticeable improvement in exercise confidence. The MCEP broadened their awareness and perceived capability towards different forms of exercise '*We thought we couldn't do* [the exercises] *and now we can do everything. . ..*' (F2 FG3) It also increased their confidence to self-monitor exercise intensity.

'*More confidence anyway in yourself you know–you were afraid to do anything in case you were doing too much or too little, you know, so you have that confidence that you know you're able to do a lot more.*' (F2 FG3)

For many this sense of empowerment extended to exercising independently. After just three months many felt they had now overcome their initial fears of unsupervised exercise '*my daughter has a treadmill at home, and I was afraid to go on it, but I go on it now you know.*' (F2 FG4) and '*Before I wouldn't know what exercises to do or anything like that and it's great to be able to do them at home, now that I understand what I should and shouldn't do.*' (F1 FG4)

**Life beyond illness.** The MCEP gave participants something to '*look forward to*' and provided many benefits that had an impact on their day-to-day living, and they wanted to continue this positive trajectory. There was a sense of 'moving on' from their illness. They talked about the programme/instructors making them feel normal again and that they were no longer defined by their condition. '*.. they don't treat us as recovering patients, and you're no longer a patient. That's a huge thing . . .. And it's by being targeted normally that you're well able to do this.*' (M3 FG 4)

Their confidence, courage, energy, and outlook on life had improved in just 10 weeks and there was a realisation that '*. . .the illness isn't the end of the road. . .*' (M2 FG3)

### Drivers of engagement

This theme contains three sub-themes; Importance of scheduled exercise; Social connections; and Enjoyment. These cover a variety of factors that motivated patients to engage with the MCEP and appeared to facilitate adherence.

**Importance of scheduled exercise.** Many lacked the motivation to exercise, admitting they were not inclined to exercise by themselves: *'you will not do it at home, you will not do it by yourself.' (P1 FG1)* The MCEP created an opportunity to maintain their exercise beyond the hospital setting.

*'You see what happens is you go and have heart problems and you get it sorted out and you go to the cardiac rehab up in Sligo hospital, which is very good. You get hooked up to machines and everything but then you get sent home and you're told carry on walking with this and within a month you go back to your normal self and you're not doing anything.' (P1 FG1)*

The scheduled nature of the MCEP was deemed important for adherence and fostered commitment: *"You know you had two dates in the week you had to meet and otherwise you might have done nothing.'* (P8, Int. 1)

**Social connections.** The social connections among the participants appeared to motivate engagement with the MCEP and, for some, started before they ever entered the programme. Although many engaged with the programme initially because they were referred through the hospital, some were also influenced by recommendations from other participants *'anyone I talked to recommended it highly.'* (F1, FG4) and *'I heard about this from my sister-in-law so I rang up to see could I join* up.' (M3 FG3) They appeared to value their recommendation and felt it encouraged their attendance at the programme.

Once attending participants particularly enjoyed the social nature of the programme. It encouraged continued participation and was one of the main drivers of exercise engagement. Participants described the sense of a collective reason for being there. Be it a similar condition to their own or another chronic condition, they felt *'everyone's in the same boat'* (M2 FG4) and *'all on the same wavelength'* (F1 FG4). This made participants *'more comfortable'* (P4 FG1) being part of the programme and helped them form social bonds. *'I felt this was great because we bounce off each other, meet people with the same situations that we have all been through and psychologically it was a chance to meet other people, talk…'* (M1 FG3) and *'you come down here and you think everyone has had stents put in but it's not, people have had different problems and you just start talking to people and it makes people at ease more…'* (M4 FG3)

There was evidence of social comparison influencing beliefs in their ability. Some felt *'put at ease'* by comparing their medical history with other participants and some viewed it rather light-heartedly. *'I'm not as bad as I thought I was* [laughing] *I've only 2 stents…'* (M3 FG3) and *'…we were counting who had the more stents at this stage you know we found out there's always someone better or worse than you'* (M1, FG3). This social comparison gave a sense that if others with a similar or even worse condition to themselves could exercise then they could too. *'…well, if they can do it so can I.'* (P2 FG1)

Participants compared themselves to new people entering the programme and realised their own ability had improved from when they started. They also talked about offering support to new people, encouraging them to stick with it and they would gradually progress.

*'You see new people coming in now and they aren't able to do what we can do, and we were once them…. we see them kind of struggling and we say don't try to do what we are doing because you won't be able for it …'* (M1 FG3)

The main component of the MCEP that fostered the social support was the post-class cup of tea *'it's the social gathering as well.'* (M1 FG3) It provided an opportunity for conversations about their medical conditions and other unrelated topics *'. . .the chat can be about anything.'* (M2 FG4)

The importance of the social support and the camaraderie they got from each other was evident from all participants *'It was the motivational encouragement you got from others* [in the group]*'* (FG1 P3) to the extent that one described it the *'centre piece of this whole thing.'* (M3 FG4)

**Enjoyment.**   Enjoyment in the programme appeared to foster motivation and exercise engagement. Most participants talked about the exercise as being 'enjoyable' and 'fun', some were even surprised they found it fun. *'We had fun too, a lot of laughter and that's very important too'* (P3 FG1). The circuits session, in particular, was associated with fun *'The wit and the banter that goes on–makes it for everybody.'* (M2 FG3)

Participants enjoyed exercising to music and found it motivating. Music added meaning and made the exercise easier and more interesting *'. . . try to do an exercise and no music, it's totally different, meaningless.'* (F2 FG3)

Many expressed a preference for more gym sessions *'I thought the gym was the best'* (P12 FG2) indicating they *'found the gym more challenging'* (F2 FG3). Others preferred the circuits *'I liked all the different exercises, and we were not long doing them. . .. The time went around quicker.'* (P6 FG1) while others liked the combination of both. Essentially a desire for variety was evident *'vary it and give you a bit more interest, you know. . ..'* (P2 FG1) with some even suggesting new alternatives such as making use of the athletics track or jiving class.

## Challenges to keeping it (exercise) up

Participants encountered challenges to keeping up the exercise. Subthemes included 'Barriers' and 'Dependency'.

**Barriers.**   The primary barrier to attending the programme was the classes times clashing with family *'clashed with dropping off kids and grandkids'* (F1, FG4) and work *'I hadn't the time to do it all the time because I was working'* (P10, FG2) commitments. Although most preferred an early morning time slot, there was concerns expressed about the dark winter mornings and the time one would have to get up if they had any distance to travel.

**Dependency.**   A key enabler of the MCEP was the availability of medical oversight and for most that was enough, for others there was a risk it was breeding medical dependency. For the first inducted group, medical personnel from the hospital were always on site and valued by participants *'It* [medical support] *was automatically there, we had the support, and it was always at the back of your mind, they are there and that is great.'* (P2, FG1)

From the second inducted group onwards, the medical personnel were only on site on certain weeks. Participants seemed to miss the reassurance from the medical side *'I thought that was very good to have somebody professional like herself here all the time, I know with staff shortages but if for future references that we could have someone here professional all the time'* (M1 FG3) with them looking for the medical staff *'regularly to be here.'* (M2 FG3)

The monitoring before and during the classes provided further reassurance of a safe exercise environment. Many noted the opportunity to have their blood pressure monitored as part of the pre-exercise health check giving them confidence *'It reassures you that you're ok for it'* (F1, FG4) or getting their pulse check during the class through random heart rate measurements to ensure they were training in the correct training zone *'you can actually see them going around to each individual and they pick out somebody who's under stress and bring them out and measure their heart rate.'* (M3 FG4) While participants saw this as a positive addition to

the programme it could indicate an overmedicalisation of exercise and create a dependency on the instructors/programme *'Getting your blood pressure taken . . . it keeps you focused on it, otherwise when would you have it taken. . .. it's nice to know you're plodding along nicely.'* (F4, FG3)

Despite the increase in exercise self-efficacy evident for many, it appears others may not exercise by themselves, some went further expressing concern in relation to losing the support of the programme. One participant feared what would happen if the programme ceased and the impact on their health. '*I hope it continues this year so that I don't land up in hospital.'* (P6, FG1) While another person referred to when there was a break in the programme, how they really missed it and felt themselves *'slip back.'* (M3 FG4)

## Discussion

The present study aimed to explore cardiac patients' experience of the early transition to and participation in a MCEP, whilst also identifying dimensions that facilitated and hindered physical activity engagement. Three key themes emerged: *moving from fear to confidence*, *drivers of engagement*, and *challenges to keeping it (exercise) up*. A key finding was that patients experienced a transition from *fear to confidence* in the early weeks attending a MCEP. Fear of exercise in individuals following a cardiac event has been found in previous studies [39] and reported as a barrier to initiating independent exercise. Despite understanding that they should have continued to exercise following the hospital-based CR, participants were fearful and uncertain of how to undertake exercise and were fearful of exercising independently. This could imply they were not taught to be independently physically active upon leaving CR or that they didn't want to exercise independently. This supports the role of a long-term step-down programme, such as a MCEP, from the hospital setting with less involvement of HCPs. For most the MCEP appeared to reduce fear and develop confidence to exercise in class and independently.

The link between the MCEP and hospital gave participant's confidence in transitioning to the MCEP, and they felt it set the MCEP apart from a regular gym or community exercise class. Participants viewed the MCEP as a safe environment to exercise. The importance of the link between community and hospital setting was evident in previous research to ensure smooth transition into the community setting [18] and establishing this link has been suggested as a measure to better support this transition [14].

Notwithstanding the benefits of continuity and reassurance highlighted by participants attending the MCEP, there is some concern that the MCEP may be over-medicalising exercise (e.g., through regular BP & HR monitoring) and not supporting a transition to more independent physical activity and long-term exercise engagement. This could breed dependency on such supervision and monitoring. For some there were signs that the dependency had just moved from the hospital to the community setting. As the MCEP programme is delivered for 60 min twice a week, participants are unlikely to achieve the recommended levels of aerobic exercise (at least 150-300min of moderate intensity aerobic exercise per week) [40]. It is a vehicle to assist them achieving the recommended two or more days per week of muscle -strengthening and functional balance activities [40]. A low dose of prescribed physical activity has been shown to be insufficient to provide meaningful benefits [41] there is an opportunity to encourage, educate, and empower the participants to increase their activity outside of class times that should be exploited. Home-based sessions incorporating telehealth monitoring has potential to support participants meet their physical activity guidelines improving long-term health and wellness [42]. Future research should assess the effect of telehealth monitoring compared to and combined with a MCEP.

The transition to the community setting can remove some of the sense of exercise as a medical treatment. Participants in the present study went so far as to saying the programme made them feel 'normal' again. This is consistent with McNamara *et al* [13] who found that exercising in a community setting promoted a sense of normality to the exercise environment. They no longer identified themselves as patients but as participants in an exercise class.

Consistent with research focusing on long-term exercise maintenance for those with a cardiac condition, this study found that scheduled exercise; social connections; and enjoyment were motivators for exercise engagement. This provision of routine and structure has been found as a key enabler to maintain exercise in previous research [18, 19]. Martin and Woods [18] reported how participants protected their class times by never scheduling other commitments that would prevent them from attending. Exercise was part of their weekly routine. Hardcastle *et al* [19] also found the discipline and routine as important with participants believing the group would be expecting them to come resulting in a 'sense of duty' to attend.

Social support from fellow participants has been consistently identified as a key driver in exercise maintenance [16–19]. The present study highlights that this social aspect can be developed very early in a programme and is fostered through the 'cup of tea and a chat'. People felt more comfortable exercising alongside others '*in the same boat*'; a phrase reported in other studies [18, 19].

A novel contribution of the present study to the literature is the evident harnessing of social comparison, whereby participants appeared to take comfort in and confidence from knowing they were not in the worst position. They were comparing their condition and exercise ability to others in the group, and this appeared to influence their perception of their ability to exercise. The behavioural change technique (BCT) of 'facilitate social comparison' has rarely been used in exercise interventions, and usually involves explicitly drawing attention to others' performance to elicit comparisons. Williams and French [43] found higher physical activity effect sizes were achieved when interventions included this BCT along with five other BCTs (i.e., provide information on consequences of the behaviour, action planning, reinforcing effort or progress towards behaviour, provide instruction, and time management). It may be the case that social comparison works in tandem with modelling to increase self-efficacy in terms of observing similar individuals and favorably comparing one's performance to that of others.

Enjoyment was another driver of exercise engagement that is consistent with previous research [17, 19]. Like Thow *et al* [17], participants expressed surprise at finding exercise fun indicating past experiences led them to believe exercise was not an enjoyable experience. Preferences differed and appealing to everyone and achieving the right balance is a challenge to the instructors. Music added to the enjoyment and was described as a key motivational aspect of the class, which has previously been noted as an important aspect of programme design in group exercise classes for older people [44].

Research carried out by Killingback *et al* [44] reported older people and individuals with CD expressed a preference for a non-gym environment. They saw gyms as boring or isolating. Morgan *et al* [45] carried out a systematic literature review on barriers to exercise referral schemes, which are similar to MCEP, and reported the gym as an intimidating environment unless activities were scheduled during off-peak hours. Previous studies appeared to use the gym for individual exercise prescription while other members where present. In the present study, participants were introduced to the gym in their group, and many expressed a preference for the gym, indicating that maybe if there were gym hours specific for the older adult it would encourage greater enjoyment and adherence to gym training. There was evidence that some participants had planned to go independently to the gym with others they had met on the programme indicating the MCEP may have assisted them in the transition to a gym environment.

Participants identified family and work commitments as key barriers to continued participation in the programme. Similar findings have been observed previously [16, 20]. Although distance was not a barrier for those in the focus group (who had completed 10 weeks), it was mentioned as a reason some of the other participants had dropped out earlier in the programme. Barriers such as cost and other health problems, which have been noted in other studies [16, 20, 39], did not appear to hinder PA engagement. This may indicate those still attending at 10 weeks were maybe physically more able having undergone hospital-based CR and better off financially. However, interviews with those who dropped out would provide further insights.

## Limitations

All participants in this study had completed the first 10 weeks of the MCEP with full intentions of continuing and may be a somewhat biased sample. The present study did not include those that had dropped out. Participants were those from a single MCEP in Sligo, Ireland, and our findings may therefore not be generalizable to other programmes. Finally, despite participants being informed that their responses would be anonymous, they were aware that these results would be seen by programme coordinators, and this may have further biased their responses.

## Strengths

The MCEP was a service as opposed to an exercise programme designed specifically for research purposes only and so this study explored implementation of an exercise programme in a 'real world' setting.

## Conclusions

The overarching theme was a transition from fear to exercise confidence following participation in the MCEP. The predominant drivers of exercise engagement were social support, enjoyment, and routine. A novel finding that emerged from this study is that participation in the MCEP in the early stages could be viewed as a double-edged sword. Undoubtedly the programme provided an exercise outlet to encourage continued exercise beyond the hospital setting, however there were signs participants were dependant on the exercise programme and were less likely to exercise outside of it that could hinder their ability to achieve PA guidelines. A further novel finding was the evident use of social comparison to provide favourable valuations of performance and increased exercise confidence. Future interventions that reduce the medicalisation of exercise, actively encourage, educate, and facilitate participants to exercise outside of the MCEP using evidence-based behavioural change techniques including facilitating social comparison would be worthwhile. The study indicates that a MCEP has potential utility in providing an exercise outlet for cardiac patients to continue to exercise following hospital-based CR, making this a sustainable model long term.

## Supporting information

**S1 Appendix. Standards for reporting qualitative research (SRQR).**
(DOCX)

**S2 Appendix. Topic guide followed by moderator in each focus group.**
(DOCX)

**S3 Appendix. Themes, subthemes and supporting quotes.**
(DOCX)

## Acknowledgments

We would like to acknowledge all study participants for their time and contribution and all research moderators who conducted the focus groups. The MCEP was based on the MedEx/ExWell Medical programme and thank Noel McCaffery and his team for supporting this project and the clinical exercise instructors in delivering the MCEP.

## Author Contributions

**Conceptualization:** Joanne Regan-Moriarty, Azura Youell, Andrew McCarren, Niall Moyna, Brona Kehoe.

**Formal analysis:** Joanne Regan-Moriarty, Sarah Hardcastle, Maire McCallion, Brona Kehoe.

**Methodology:** Joanne Regan-Moriarty, Maire McCallion, Azura Youell, Niall Moyna, Brona Kehoe.

**Resources:** Maire McCallion, Azura Youell, Audrey Collery.

**Supervision:** Andrew McCarren, Niall Moyna, Brona Kehoe.

**Validation:** Sarah Hardcastle.

**Writing – original draft:** Joanne Regan-Moriarty.

**Writing – review & editing:** Joanne Regan-Moriarty, Sarah Hardcastle, Maire McCallion, Azura Youell, Audrey Collery, Andrew McCarren, Niall Moyna, Brona Kehoe.

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
