## [Decision Letter · Decision Letter 0]

21 Dec 2023

PONE-D-23-24657‘The illness isn’t the end of the road’ - Patient perspectives on the initiation of and early participation in a multi-disease, community-based exercise programmePLOS ONE

Dear Dr. Moyna,

Thank you for submitting your manuscript to PLOS ONE. After careful consideration, we feel that it has merit but does not fully meet PLOS ONE’s publication criteria as it currently stands. Therefore, we invite you to submit a revised version of the manuscript that addresses the points raised during the review process.

We look forward to receiving your revised manuscript.

Kind regards,

Michael Thomas Lawless, Ph.D.

Academic Editor

PLOS ONE

Journal Requirements:

4. We note that your Data Availability Statement is currently as follows: [Add Data Availability statement here]

Reviewers' comments:

Reviewer's Responses to Questions

**Comments to the Author**

1. Is the manuscript technically sound, and do the data support the conclusions?

Reviewer #1: Yes

Reviewer #2: Yes

2. Has the statistical analysis been performed appropriately and rigorously? 

Reviewer #1: N/A

Reviewer #2: Yes

3. Have the authors made all data underlying the findings in their manuscript fully available?

Reviewer #1: Yes

Reviewer #2: Yes

4. Is the manuscript presented in an intelligible fashion and written in standard English?

Reviewer #1: Yes

Reviewer #2: Yes

5. Review Comments to the Author

Reviewer #1: This manuscript about patient perspectives of a community based exercise programme for people with chronic health conditions is very well written.

I wonder if the following could be considered?

If data saturation was considered when determining the number of study participants?

If transcripts were returned to participants and/or if participants checked or provided comment on the themes?

If software was used to analyse the data (such as Nvivo)?

The results section is very clear although I did wonder why some quotes were part of a sentence and others were indented and included on the next available line?

Line 416 wasn't should be was not.

I did also wonder if the exercise classes were going to be continued or not? and if this could be reported.

Thank you.

Reviewer #2: The manuscript is well-written in an engaging and lively style. It’s currently something of interesting topic and the purpose of this study was to investigate the patients' experiences of the initiation and early participation in a MCEP CR programme. Some areas require rewriting, clarification or editing. I comment on these areas section:

Line 72-75: Please say what is the main barriers that only less than half receive CR? Specifically for women?

(ie. REFS: https://doi.org/10.1016/j.cjca.2023.07.016 ; https://doi.org/10.3390/ijerph182413113 ; https://doi.org/10.3390/ijerph20054064

Methodology:

Should be improved and add more details

-Please define the way of samples’ selection. Was it random? Please describe the way of choosing them. How the sample size was defined?

-Please say more about CR programme. Describe FITT parameters. What about those who was taking b-blockers?

-I suggest to add a table with CR components.

-What are the primary and secondary outcomes?

-Please add a flowchart.

Discussion

What about safety? Does it affect participation?

Please add a short discussion what would be the ideal dose of CR (how many weeks are enough?). There are several hundred references in the field of CR, in the last years. doi: 10.1136/heartjnl-2012-303055

Please

Future research:

include in the limitations the strengths of this study, a justification that the search missed the field of telehealth interventions primarily based on a home-based approach and include keywords (telehealth, mhealth, telerehab).

doi:10.7759/cureus.23485

https://doi.org/10.1016/j.jbmt.2021.07.009

DOI:https://doi.org/10.1016/j.ijcard.2022.08.055

6. PLOS authors have the option to publish the peer review history of their article (what does this mean?). If published, this will include your full peer review and any attached files.

Reviewer #1: **Yes: **Dr Leica Sarah Claydon-Mueller

Reviewer #2: No

---

## [Author Response · Author response to Decision Letter 0]

11 Feb 2024

Feb 2024

Thank you for reviewing and forwarding feedback on the manuscript. We appreciate the opportunity to submit a revised draft of the manuscript for consideration. Please find included below responses to each point raised by the academic editors and reviewer(s). The following files have been uploaded. 

1. 'Response to Reviewers' - within this Rebuttal letter we have responded to specific reviewer and editor comments 

2. ‘Revised Manuscript with Track Changes’

2. ‘Manuscript’ - an unmarked version

3. ‘Fig1.tif’ - image of flowchart requested

4. ‘S1_Appendix’ – file renamed

5. ‘S2_Appendix’ – file renamed 

6. S3_Appendix – updated and renamed

We hope you find the responses to all comments within the rebuttal letter meet your satisfaction. Please contact us if you have any further recommendations or amendments you require. 

Sincerely

Dr. Joanne Regan-Moriarty,

Department of Health and Nutritional Science, ATU Sligo, e-mail: joanne.regan@atu.ie

Corresponding Author

Prof Niall Moyna, DCU, Dept of Health and Human Performance, Dublin, e-mail : niall.moyna@dcu.ie

---

## [Editor Report · Decision Letter 1]

16 Feb 2024

‘The illness isn’t the end of the road’ - Patient perspectives on the initiation of and early participation in a multi-disease, community-based exercise programme

PONE-D-23-24657R1

Dear Dr. Moyna,

We’re pleased to inform you that your manuscript has been judged scientifically suitable for publication and will be formally accepted for publication once it meets all outstanding technical requirements.

Kind regards,

Michael Thomas Lawless, Ph.D.

Academic Editor

PLOS ONE
---

## [Editor Report · Acceptance letter]

20 Mar 2024

PONE-D-23-24657R1 

PLOS ONE

Dear Dr. Moyna, 

I'm pleased to inform you that your manuscript has been deemed suitable for publication in PLOS ONE. Congratulations! Your manuscript is now being handed over to our production team.

Kind regards, 

on behalf of

Dr. Michael Thomas Lawless 

Academic Editor

PLOS ONE